# Establishment of a UPLC-PDA/ESI-Q-TOF/MS-Based Approach for the Simultaneous Analysis of Multiple Phenolic Compounds in Amaranth (*A*. *cruentus* and *A. tricolor*)

**DOI:** 10.3390/molecules25235674

**Published:** 2020-12-01

**Authors:** Won Tea Jeong, Jun-Hyoung Bang, Seahee Han, Tae Kyung Hyun, Hyunwoo Cho, Heung Bin Lim, Jong-Wook Chung

**Affiliations:** Department of Industrial Plant Science and Technology, Chungbuk National University, Chungbuk, Cheongju 28644, Korea; shewaspretty@chungbuk.ac.kr (W.T.J.); peerage8794@gmail.com (J.-H.B.); saehee@gmail.com (S.H.); taekyung7708@chungbuk.ac.kr (T.K.H.); hwcho@chungbuk.ac.kr (H.C.)

**Keywords:** UPLC-PDA, ESI-Q-TOF/MS, phenolic compounds, *Amaranthus* species

## Abstract

We used ultraperformance liquid chromatography coupled with a photodiode-array detector and electrospray ionization quadrupole time-of-flight mass spectrometry (UPLC-PDA/ESI-Q-TOF/MS) to rapidly and accurately quantify 17 phenolic compounds. Then, we applied this method to the seed and leaf extracts of two *Amaranthus* species to identify and quantify phenolic compounds other than the 17 compounds mentioned above. Compounds were eluted within 30 min on a C18 column using a mobile phase (water and acetonitrile) containing 0.1% formic acid, and the specific wavelength and ion information of the compounds obtained by PDA and ESI-Q-TOF/MS were confirmed. The proposed method showed good linearity (r^2^ > 0.990). Limits of detection and quantification were less than 0.1 and 0.1 μg/mL, respectively. Intra- and interday precision were less than 2.4% and 1.8%, respectively. Analysis of amaranth seed and leaf extracts using the established method showed that the seeds contained high amounts of 2,4-dihydroxybenzoic acid and kaempferol, and leaves contained diverse phenolic compounds. In addition, six tentatively new phenolic compounds were identified. Moreover, seeds potentially contained 2,3-dihydroxybenzaldehyde, a beneficial bioactive compound. Thus, our method was an efficient approach for the qualitative and quantitative analysis of phenolic compounds, and could be used to investigate phenolic compounds in plants.

## 1. Introduction

Plant phenolic compounds, a group of secondary metabolites, are largely classified as flavonoids or nonflavonoids depending on the number of bound phenolic structures [1,2]. Phenolic compounds are potent antioxidants that inhibit reactive oxygen species (ROS)-related enzymes such as xanthine oxidase, cyclo-oxygenase (COX), lipoxygenase, and phosphoinositide 3-kinase [3,4,5]. In addition, since phenolic compounds reduce inflammation, and treat Alzheimer’s disease and arteriosclerosis, the consumption of plants rich in these compounds is encouraged to prevent the onset of various diseases and aging [6].

Although several fruits and vegetables are rich in phenolic compounds, the search for grains and medicinal plants containing large amounts of phenolic compounds continues [7]. The *Amaranthus* genus comprises approximately 70 dicotyledonous plant species collectively known as amaranths, of which 17 are used as dual-purpose crops for seeds and leafy greens, and 3 are consumed for their seeds [7]. Amaranth leaves and seeds contain high amounts of proteins, vitamins, minerals, and phenolic compounds; extracts of some species, such as *Amaranthus tricolor*, are recognized as health foods since they inhibit the growth of human tumor cells [8]. Amaranths show a variety of leaf and seed colors. Amaranth leaves vary in color from red to green, and the *Amaranthus cruentus* species contains a high content of beta-alanine, which shows strong antioxidant activity [9].

Various phenolic compounds with high antioxidant activity also exist in amaranths with green leaves and seeds; therefore, these plants can be a good source of natural antioxidants [10,11]. However, while the distribution of phenolic compounds in *Amaranthus* species varies with leaf color, the content of phenolic compounds is affected by environmental factors, i.e., the same species may contain different amounts of phenolic compounds under different environmental conditions. Therefore, an analytical approach is required for the qualitative and quantitative analysis of the distribution of phenolic compounds in amaranths [12].

Phenolic compounds are frequently quantified using either the colorimetric method (for measuring total phenol and total flavonoid contents), or by liquid chromatography coupled with an ultraviolet detector (LC–UVD) or mass spectrometry (LC–MS) [8,13]. The colorimetric method is simple and convenient, but cannot determine individual compounds; therefore, LC is used when the objective is to accurately identify compounds in complex samples. LC–UVD and LC, followed by tandem MS (LC–MS/MS), are used to separate individual compounds, and because these methods have excellent selectivity and sensitivity, trace quantitation is possible [14]. However, a disadvantage of these methods is that a standard substance must be retained to identify and quantify individual compounds. Electrospray ionization quadrupole time-of-flight MS (ESI-Q-TOF/MS), a commercial high-resolution mass spectrometry method, is recognized as an excellent platform for profiling various metabolites such as phenolic compounds, alkaloids, and terpenoids because of its ability to identify and quantify unknown compounds without the need for a standard [15]. However, since ESI-Q-TOF/MS cannot accurately quantify some phenolic compounds, analysis is performed by LC–UVD for the simultaneous analysis of other phenolic compounds.

In this study, we used ultraperformance liquid chromatography coupled with a photodiode-array detector and ESI-Q-TOF/MS (UPLC-PDA/ESI-Q-TOF/MS) to quickly and accurately identify 17 phenolic compounds. This method was applied to the leaf and seed extracts of two different *Amaranthus* species collected from various countries. Furthermore, this method was used to identify other phenolic substances by providing wavelength and mass information to easily identify phenolic compounds, and to perform the chemical profiling of unknown peaks. To the best of our knowledge, this was the first attempt to investigate phenolic compounds in *Amaranthus* using high-resolution such as Q-TOF (Figure 1).

## 2. Results and Discussion

### 2.1. UPLC-PDA/ESI-Q-TOF/MS

Figure 2 shows the chromatogram obtained using the PDA for compounds separated by UPLC. An aqueous solution containing formic acid and acetonitrile was optimized using a C18 column under the conditions of Section 3.2, and separation into individual peaks was achieved within 23 min. In several previous studies, separation of phenolic compounds by high-performance liquid chromatography (HPLC)-ESI-Q-TOF/MS took at least 40 min on average; however, in this study, UPLC decreased separation time, and the obtained peaks showed excellent resolution [16,17,18].

Unfortunately, vanillic acid, syringic acid, benzoic acid, and sinapinic acid were only detectable using the PDA. These results were assumed to be affected by the pH of the mobile phase. To enable stable data acquisition, the pH of the aqueous solution used as the mobile phase for the LC–UVD analysis of phenolic and flavonoid compounds is generally adjusted with formic acid and sometimes with phosphoric acid [19,20].

Although the above material was analyzed by LC–MS/MS using an aqueous mobile phase containing formic acid in some studies, no study has reported the use of the TOF approach to perform multicomponent analysis. To analyze benzoic acid by LC–MS, the pH of the mobile phase was adjusted to more than 4.5 with ammonium acetate or ammonium carbonate [21,22,23]. The use of ammonium acetate increases the retention time of each component, which increases analysis time and lowers overall sensitivity, and this is not suitable for the simultaneous analysis of multiple components. Therefore, the use of an aqueous mobile phase with formic acid added was ideal for the simultaneous analysis of phenolic compounds. The combined use of UVD and Q-TOF serves as an optimal tool for overcoming these shortcomings, and is preferred by many researchers for analyzing phenolic compounds [24].

Table 1 shows the maximal absorption wavelength and characteristic mass information of the peaks corresponding to each compound separated by UPLC. UPLC-PDA analysis was performed at wavelength settings of 280, 254, and 310 nm, and peak intensity varied with wavelength. Overall absorbance was good at 280 nm. Maximal absorption wavelength (Table 1) and mass information can accurately identify the compound. ESI-Q-TOF analysis was performed in negative ion mode, and all detected compounds were ionized with quasimolecular ions, [M − H]^−^. The best collision energy for observing prominent ions among collision-induced dissociation (CID)-derived fragmented ions was 30–40 V.

Phenolic compounds are ionized in both positive and negative modes, but the prediction of ionized mass value and fragmentation is better in negative mode [24,25,26]. In addition, when performing chemical profiling, negative mode is advantageous for the potential identification of phenolic compounds because compounds containing nitrogen with an unshared electron pair are not detected [24]. Therefore, positive mode was not used in this study. This wavelength and mass information can be used as a library to accurately identify the compound and provide important clues for profiling.

### 2.2. Method Validation

Table 1 shows the verification results of the UPLC-PDA method. The calibration curves of all standards displayed good linearity, with the coefficient of determination (r^2^) greater than 0.990. The limits of detection and quantification were within ranges of 15–50 and 60–150 ng/mL, respectively. Intra- and interday precision for the analytes under UPLC-PDA conditions was less than 1.81% and 1.55%, respectively. The RSD range of the precision required by ICH is 4–6%, and our method showed excellent precision, not exceeding 2%. Accuracy was confirmed in the range of 93.4–106.1%.

### 2.3. Chemical Profiling of Phenolic Compounds in Seed Samples

Chemical profiling was performed to confirm the presence of phenolic compounds other than those detected by comparing their retention time (*t*_R_), absorption wavelength, and fragment ions with those of the standard. Chemical profiling uses UNIFI software to calculate the mass value of the parent molecule corresponding to the unknown peak with high resolution, excluding the peak matched with the standard, and calculates the expected molecular formula of the extracted phenolic compounds containing carbon, hydrogen, and oxygen. The molecular formula was determined, so that [M − H]^−^ quasimolecular ions were established within 30 ppm of mass error, and further investigated with ChemSpider and through literature searches. Results showed peaks that presumably represented a total of 6 phenolic compounds (Table 2). No previous reports suggested that the identified compounds were found in *Amaranthus*, and the peak with *m*/*z* 137 was found in all seed samples. This ion was tentatively identified as two candidates (salicylic acid and 2,3-dihydroxybenzaldehyde) that could exist in plants through chemical profiling. Although this ion showed the same mass ion of the parent molecule as salicylic acid, *m*/*z* 93, a representative fragment ion produced by CID from the parent molecule of salicylic acid was not observed in our results [27,28].

To ensure accurate identification, it is important to check the conformity of various factors using standard materials. Compound 2,3-dihydroxybenzaldehyde exhibited strong antioxidant activity, protecting against hypotensive effects, genotoxicity, and cytotoxicity [29,30]. In order to receive attention as a valuable food along with the leaves of *Amaranthus*, it is necessary to try more elaborate confirmation by comparing retention time, mass ions, and fragmentation ions after securing the real standard of the substance.

### 2.4. Heat Map and Clustering Analysis

Heat maps for the 17 phenolic compounds identified in the seeds and leaves of 7 *Amaranthus* plant sources are shown in Figure 3 (green indicates a higher content than the average value). Clustering analysis divided the phenolic compounds into two categories. All phenolic compounds identified in the leaves except 3,4-dihydroxybenzoic acid and kaempferol were dispersed, and no clear difference was apparent among *Amaranthus* species.

### 2.5. Assessing the Greenness

On the other hand, in the seeds, 3,4-dihydroxybenzoic acid was detected at a high level in the *A. cruentus* collected from two countries, suggesting that 3,4-dihydroxybenzoic acid was unaffected by environmental factors. The kaempferol content was higher than that of other phenolic compounds in ACS1, ATS3, and ATS4, although this result was presumably not related to the difference in content between resources. However, these two components exhibited common features that were not detected in mature leaves. According to previous studies, the content of benzoic acid and flavonoids isolated from the same plant species may differ between regions, and total phenol and flavonoid contents were higher in leaves than in seeds [12,31]. In addition, amaranth leaves show high antioxidant activity [12]. The difference in the content of phenolic compounds between seeds and leaves was consistent with the results of the existing literature. However, the difference in the contents of benzoic acid and kaempferol between leaves and seeds has not been reported to date. In contrast to our results, Hongyan et al. showed that *Amaranthus caudatus* contained a higher content of several phenolic compounds (2,4-dihydroxybenzoic and kaempferol-3-rutinoside) including 3,4-dihydroxybenzoic acid in leaves than in seeds [31]. On the other hand, Magdalena Karamac et al. measured the phenol content of *A. caudatus* from the early vegetative stage to the grain stage, and reported that the contents of caffeic acid and hydroxycinnamic acid derivatives decreased [8]. It is important to examine whether this difference represents the difference between resources or the effect of environmental factors. In addition, it is necessary to investigate contents of kaempferol in various resources.

Ecoscale evaluation was performed to evaluate whether our analytical method was suitable for greenness [32]. Ecoscale is based on assigning penalty points (PPs) to parameters used from sample preparation to analysis by Gu et al. [23]. Among recently reported studies, three papers, similar to ours but with different analyzers, were selected to evaluate and compare rust formation. Results for penalty points are shown in Table 3—the fewer the points are, the closer to greening. This result showed that our method was closer to green formation than the methods of HPLC-UV and HPLC-ESI-Q-TOF were.

## 3. Materials and Methods

### 3.1. Chemicals and Reagents

All solvents and reagents used for analysis were HPLC- and analytical-grade. Gallic acid (≥98.0%), 3,4-dihydroxybenzoic acid (≥97.0%), 4-hydroxybenzoic acid (≥99.0%), 2,4-dihydroxybenzoic acid (≥97.0%)), vanillic acid (≥97.0%), caffeic acid (≥98.0%), syringic acid (≥95.0%), p-coumaric acid (≥98.0%), ferulic acid (≥99.0%), sinapinic acid (≥98.0%), rutin (≥95.0%) quercetin 3-β-d-glucoside (≥99.0%), benzoic acid (≥99.5%), kaempferol 3-*O*-β–rutinoside (≥98.0%), quercetin (≥95.0%), cinnamic acid (≥99.0%), and kaempferol (≥97.0%) were purchased from Sigma-Aldrich (St. Louis, MO, USA).

Stock standard solutions were prepared by dissolving approximately 1 mg of each substance in a 1 mL aliquot of methanol (1000 μg/mL). A mixed standard solution containing 10 μg/mL was subsequently prepared in methanol and stored at 4 °C prior to use.

### 3.2. Sample and Extraction

Plant materials were collected from the Rural Development Administration (RDA)-Genebank Information Center in Korea. *Amaranthus* species used in the study and source details are listed in Appendix A. Seeds were sown in April 2019, and seedlings were transplanted at a research farm at the Chungbuk National University, Korea (36°37′27.7′′ N, 127°27′3′′ E). The seedlings were cultivated from May to August 2019. Harvested samples were lyophilized, ground into a fine powder, and used for the extraction of phenolic compounds.

Fresh leaves (100 mg) were submitted to infusion in 1 mL of 70% aqueous ethanol by sonicating for 1 h, and the sample was centrifuged for 10 min at 12,000 rpm at 4 °C. The supernatant was filtered with a 0.2 μm filter, diluted 10 times with distilled water, and used for instrument analysis.

### 3.3. Analysis Conditions

The identification and quantification of the phenolic compounds was carried out with the use of a UPLC system (Waters, Milford, MA, USA) equipped with a binary solvent pump, an autosampler, and a photodiode-array detector. MS analysis was performed using a Xevo G2 Q-TOF model coupled with ESI (Waters, Milford, MA, USA). UPLC and Q-TOF conditions are summarized in Table 4. The mobile phase was composed of water (Solvent A) and acetonitrile (Solvent B) containing 0.1% formic acid. Mass-acquisition mode was performed in negative mode.

### 3.4. Method Validation

Method validation was evaluated for linearity, limit of detection (LOD), limit of quantitation (LOQ), accuracy, and precision (intra- and interday) by modifying according to International Conference on Harmonization (ICH) guidelines.

A mixture of standard solutions of all 17 phenolic compounds was prepared at 7 different concentrations (ranging from 0.05 to 10 μg/mL). Linearity was evaluated by the coefficient of determination of the regression equation. Linearity-calibration curves were constructed by the least-seven plotting of each reference compound and evaluated by a squared–linear correlation coefficient (r^2^). LOD and LOQ were defined as signal-to-noise (S/N) ratios of 3 and 9, respectively. Inter- and intraday precision were determined by analyzing for 6 replicates for known concentration (concentration in the middle of the calibration curve) on the same day (intraday) and on 3 consecutive days (interday), and expressed in terms of relative standard deviation (RSD). Accuracy was evaluated as the proportion of the standard concentration in the spiked sample analyzed and the theoretical spiked concentration.

### 3.5. Statistics

All the data were converted to the standard score. Heat-map hierarchical clustering was performed using R software (version 3.6.3, R Foundation for Statistical Computing, Vienna, Austria). The hierarchical cluster was analyzed using the ward.D method on the basis of Pearson’s correlation coefficient.

## 4. Conclusions

A method to quickly and accurately analyze 17 phenolic compounds with UPLC-PDA/ESI-Q-TOF/MS was developed and verified. This method was successfully applied to amaranths, a rich source of potent antioxidants, to quantify major phenolic compounds. Six new phenolic compounds were tentatively identified by ESI-Q-TOF. Although components detected with the PDA were included, the ability to identify unknown components of ESI-Q-TOF/MS offset this problem. Our proposed method of analysis could help researchers understand the distribution of phenolic compounds in *Amaranthus* species and identify phenolic compounds in other plant species.

## Figures and Tables

**Figure 1 molecules-25-05674-f001:**
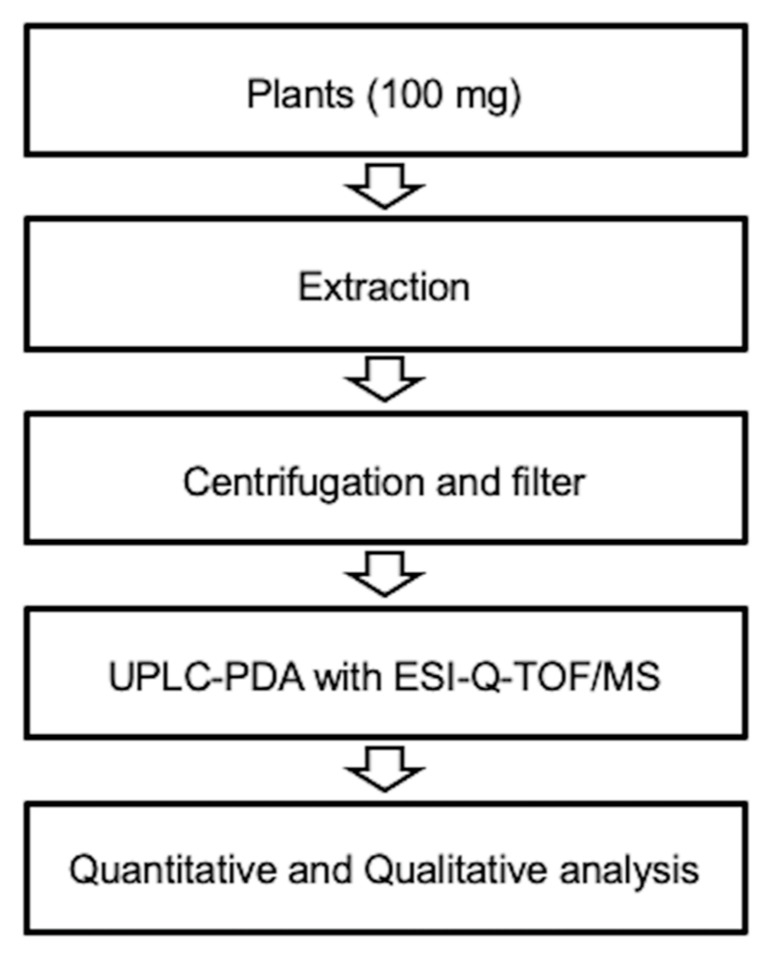
Schematic flowchart showing steps for analysis of 17 phenolic compounds in *Amaranthus* by ultraperformance liquid chromatography coupled with photodiode-array detector and electrospray ionization quadrupole time-of-flight mass spectrometry (UPLC-PDA/ESI-Q-TOF/MS).

**Figure 2 molecules-25-05674-f002:**
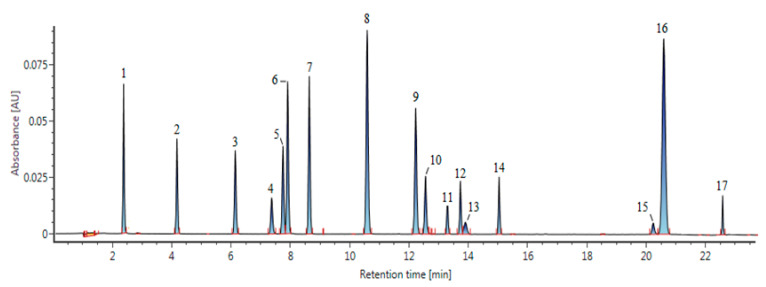
Chromatograms of standard mixture of 17 phenolic compounds at 260 nm. Peaks were identified as follows: (1) gallic acid, (2) 3,4-dihydroxybenzoic acid, (3) 4-hydroxybenzoic acid, (4) 2,4-dihydroxybenzoic acid, (5) vanillic acid, (6) caffeic acid, (7) syringic acid, (8) p-coumaric acid, (9) ferulic acid, (10) sinapinic acid, (11) rutin, (12) quercetin 3-β-d-glucoside, (13) benzoic acid, (14) kaempferol 3-*O*-β-glucoside, (15) quercetin, (16) cinnamic acid, (17) kaempferol.

**Figure 3 molecules-25-05674-f003:**
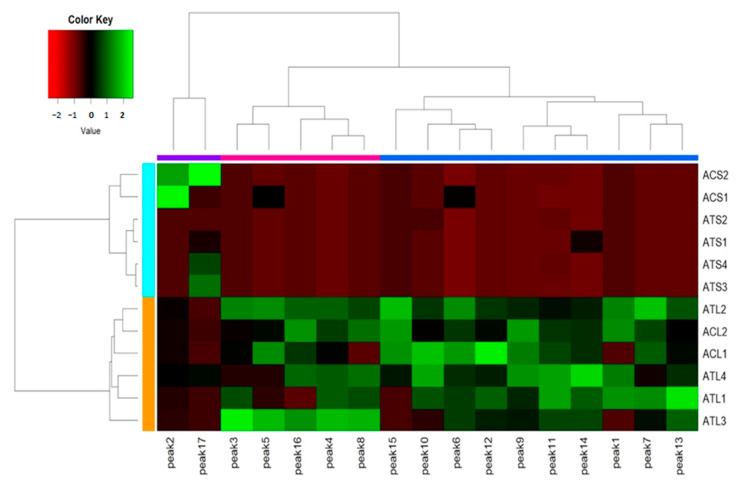
Heat map of contents of phenolic compounds in *Amaranthus* species.

**Table 1 molecules-25-05674-t001:** Results of method validation and characterization of mass information for 17 phenolic compounds by UPLC-PDA/ESI-Q-TOF/MS. Note: LOD, limit of detection; LOQ, limit of quantification.

Peak No.	Compound Name	PDA Detector								ESI-Q-TOF/MS			
		*t*_R_ (min)	r^2^	λ_max_	LOD (µg/mL)	LOQ (µg/mL)	Precision (RSD, %)		Accuracy (%)	Theoretical *m*/*z*	Observed *m*/*z*	Mass Error (ppm)	Major Fragment Ions (*m*/*z*)
							Intraday	Interday					
1	Gallic acid	2.35	0.999	270.4	0.01	0.03	2.4	1.8	104.3	170.0215	169.0084	30.5	125.0187, 123.0030
2	3,4-Dihydroxybenzoic acid	4.14	0.999	258.5/293.1	0.05	0.15	0.7	0.9	99.3	154.0266	153.0136	33.1	109.0238, 108.0160, 107.0086
3	4-Hydroxybenzoic acid	6.12	0.996	254.9	0.05	0.15	0.4	0.4	101.4	138.0317	137.0187	36.9	93.0293
4	2,4-Dihydroxybenzoic acid	7.27	0.999	253.7/293.0	0.05	0.15	1.4	1.3	98.8	154.0266	153.0135	33.7	109.0238
5	Vanillic acid	7.73	0.999	259.7/290.6	0.05	0.15	0.4	0.7	102.7	-			
6	Caffeic acid	7.89	0.999	243.1/324.1	0.05	0.15	0.8	1.0	101.6	180.0423	179.0292	28.8	135.0391, 134.0312, 133.0236
7	Syringic acid	8.63	0.999	274.0	0.01	0.03	0.4	0.5	106.1	-			
8	p-Coumaric acid	10.57	0.999	309.7	0.02	0.06	0.4	0.3	105.3	164.0473	163.0345	29.8	119.0446, 117.0291, 93.0294
9	Ferulic acid	12.21	0.999	240.7/322.9	0.01	0.03	0.1	0.3	93.4	194.0579	193.045	25.7	134.0316, 133.0236
10	Sinapinic acid	12.54	0.999	239.5/324.1	0.1	0.3	0.3	0.5	102.1	224.0685	223.0557	21.8	164.0419, 149.0186, 121.0237
11	Rutin	13.27	0.997	256.1/348.7	0.1	0.3	0.7	1.1	102.306	610.1534	609.1467	−2.0	301.0302, 300.0233
12	Quercetin 3-β-d-glucoside	13.71	0.999	254.9/348.7	0.1	0.3	0.6	0.6	96.6	464.0955	463.0862	2.9	301.0300, 300.0234
13	Benzoic acid	13.89	0.999	236.0/273.9	0.1	0.3	0.4	0.3	98.5	-			
14	Kaempferol 3-*O*-β-rutinoside	15.01	0.990	265.6/348.7	0.1	0.3	0.3	0.3	104.2	594.1585	593.1522	−2.8	285.0360, 284.0285
15	Quercetin	20.17	0.997	275.1	0.2	0.6	1.6	1.4	95.1	302.0427	301.0312	11.8	178.9928, 158.9979, 121.0237
16	Cinnamic acid	20.54	0.999	276.3	0.005	0.015	0.4	0.5	100.0	-			
17	Kaempferol	22.55	0.999	265.6/366.8	0.05	0.15	1.4	1.8	104.3	286.0477	285.0361	12.8	159.0390, 143.0444, 93.0293

**Table 2 molecules-25-05674-t002:** Characteristic mass information of compounds tentatively identified in two *Amaranthus* species by UPLC-PDA/ESI-Q-TOF/MS.

Peak no.	*t*_R_ (min)	Observed [M − H]^−^ (*m*/*z*)	Neutral Mass (Da)	Mass Error (mDa)	Formula	Tentatively Identified Compounds	Major Fragment Ions (*m*/*z*)	Detected Sample Name
1	5.98	137.0188	138.0317	−5.8	C_7_H_6_O_3_	2,3-Dihydroxybenzaldehyde	108.0165 (59.1)	All seed samples
2	6.22	385.0742	386.0849	−3.5	C_16_H_18_O_11_	*O*-Feruloylgalactaric acid	215.0271 (14.6), 212.9966 (68.7), 191.0141 (100), 147.0241 (24.2), 85.0243 (57.33)	All leave sample
3	10.08	581.1518	582.1585	0.6	C_26_H_30_O_15_	Flavonol 3-*O*-d-xylosylgalactoside	443.1181 (23.2), 351.0652 (6.4), 257.0384 (3.2), 167.0294 (75.5), 137.0188 (39.5)	All seed samples
4	11.88	755.2073	756.2113	3.3	C_33_H_40_O_20_	Kaempferol 3-(3*R*-glucosylrutinoside)-Faralateroside	555.1208 (4.9), 363.0688 (2.6), 300.0232 (24.6), 201.0115 (4.2)	All leaf samples
5	16.42	449.1464	450.1526	1.1	C_22_H_26_O_10_	Asebotin	433.1573 (1.0), 240.9976 (6.7), 177.0341 (2.9), 150.9652 (1.3)	All leaf samples
6	17.87	287.0914	288.0998	−1.1	C_16_H_16_O_5_	Shikalkin	235.0563 (1.5), 195.0608 (5.2)	ATL1, ATL2, ATS1 and ATS2

**Table 3 molecules-25-05674-t003:** Penalty points (PPs) for determination of phenolic compounds in this study and others.

	Our Method		Reference [19]		Reference [24]	
Reagent		PPs		PPs		PPs
	Ethanol 1 mL	0	NaOH (2.5–7.5 M) 4 mL	2	Ethanol	0
			Dichloromethane 2 mL	1		
Instruments						
	UPLC-UV	0	Heater	8	HPLC-UV	1
	MS(ESI-Q-TOF/MS)	2	HPLC-UV	1	MS(ESI-Q-TOF/MS)	2
	Waste	0	Waste	1	Waste	0
Total penalty points		∑2		∑13		∑3

**Table 4 molecules-25-05674-t004:** UPLC-PDA/ESI-Q-TOF/MS operating conditions.

Parameters	Conditions	
UPLC PDA conditions bellow;		
Injection volume	5 μL	
Column temperature	40 °C	
Flow rate	0.25 mL/min	
Column type	BEH C18 column (2.1 × 100 mm, 1.7 μm)	
Gradient/mobile phase	Solvent A (%)	Solvent B (%)
Time (min)		
0	98	2
20	75	25
24	40	60
27	10	90
28	10	90
30	98	2
35	re-equilibration	
ESI-Q-TOF/MS conditions bellow;		
Capillary voltage	3.0 kV	
Cone voltage	30 V	
Cone gas flow	800 L/h	
Desolvation gas flow	60 L/h	
Source temperature	40 °C	
Scan time	0.25 s	
Scan range	*m*/*z* 50–1200	
Collision energy	Low-collision energy, 6 eV; high-collision energy, 30–50 eV	
Software	UNIFI ver. 1.8	

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
