# Peer review of "Establishment of a UPLC-PDA/ESI-Q-TOF/MS-Based Approach for the Simultaneous Analysis of Multiple Phenolic Compounds in Amaranth (A. cruentus and A. tricolor)"

_molecules, 2020, doi:10.3390/molecules25235674_

Round 1

Reviewer 1 Report

This study has been conducted using cultivated samples, making a comparative study between specimens collected in their natural habitat and cultivated specimens extremely important. 

Each seed has a different genotype.  

Table 1 must be clarified 

Are the 4 samples of Amaranthus tricolor – 3 of them collected in Mexico and 1 in CMR – varieties? Were they collected during the same season and/or daytime? 

Are the 3 samples of Amaranthus cruentus – collected in GHA, FIN and unknown location – varieties? Were they collected during the same season and/or daytime? Samples with unknown origin can not be used for study. 

The paper studies only two species, which is a very small sample for comparison and informative analysis. 

Author Response

We are grateful for your invaluable comments on our manuscript. As per the reviewer’s suggestion, In addition, I have revised and added to reviewers.  Herewith, I have given our response to reviewer comments and revised the manuscript by using shadow text. I hope to get a positive answer from you soon.

This study has been conducted using cultivated samples, making a comparative study between specimens collected in their natural habitat and cultivated specimens extremely important.

Each seed has a different genotype. 

Table 1 must be clarified

Are the 4 samples of Amaranthus tricolor – 3 of them collected in Mexico and 1 in CMR – varieties? Were they collected during the same season and/or daytime?

Are the 3 samples of Amaranthus cruentus – collected in GHA, FIN and unknown location – varieties? Were they collected during the same season and/or daytime? Samples with unknown origin can not be used for study.

The paper studies only two species, which is a very small sample for comparison and informative analysis.

Thank you very much for reviewing our thesis. We tried to understand the reviewer's comments. First of all, I apologize for the fact that section 3-2 should have been written as supplmentary, but as table 1. We have corrected this part. As the reviewer said, I think the study between samples is very important. However, the purpose of our study was to establish a method that can analyze 17 phenolic compounds using two tools, and the second was whether the established analysis method could be applied to real samples. The resources used in this study were not directly collected by us.

-> Thank you very much for reviewing our thesis. We tried to understand the reviewer's comments. First of all, I apologize for the fact that section 3-2 should have been written as supplmentary, but as table 1. We have corrected this part. As the reviewer said, I think the study between samples is very important. However, the purpose of our study was to establish a method that can analyze 17 phenolic compounds using two tools, and the second was whether the established analysis method could be applied to real samples. In addition, the resources used in this study were not directly collected by us. The seeds in this study were distributed from Rural Development Administration (RDA)- Genebank in Korea, and then Amaranthus was cultivated directly by us as described in the section 3.2. We indicated the country of origin because we analyzed the phenolic compounds of the seeds as they were collected. We analyzed the phenol content, but we used a heatmap because we wanted to further study (because the two resources show differences in color with the naked eye) to see if there is a difference between species. Since we agree with the reviewer's opinion that samples of unknown origin cannot be used in the study, the samples were removed and the heatmap was reconstructed. Please consider our purpose.

Reviewer 2 Report

This paper by Korean authors deal with the development of modern analytical method for quantification of phenolic compounds in amaranthus plant and seeds. Although the investigation brings no breakthrough, it is well performed and properly designed and the language is acceptable. The paper is well-written and after minor corrections can be published.

Please consider the following corrections:

Line 86:  The compounds are described as a phenolic acids, not all of them are carboxylic acids (eg 15). Please correct.

Line 144:  There is something wrong with the sentence starting on this line. Please correct.

Table 1:  Although unsure the opponent thinks  that  the mass error is much lower than stated.  Maybe the authors did not accounted for the mass of one electron (0.00054858  Da). Please consider.

Table 2: Please use subscript in molecular formula as per usual. 

Omit the CIP notation in the tentatively identified compounds. The detection method can not in principle provide such information.

Figure 2:  The heat map is very useful. Please enlarge all all the text as is it is not easy to read.  Please correct  all the the compound names (eg some start with X). Add units to the value [mg/g dry weight?].

Line 190: Please include and properly number the table mentioned in the text with the informations. Table 1 shows completely different informations.

Author Response

We are grateful for your invaluable comments on our manuscript. As per the reviewer’s suggestion, In addition, I have revised and added to reviewers.  Herewith, I have given our response to reviewer comments and revised the manuscript by using shadow text. I hope to get a positive answer from you soon.

This paper by Korean authors deal with the development of modern analytical method for quantification of phenolic compounds in amaranthus plant and seeds. Although the investigation brings no breakthrough, it is well performed and properly designed and the language is acceptable. The paper is well-written and after minor corrections can be published.

Please consider the following corrections:

Line 86:  The compounds are described as a phenolic acids, not all of them are carboxylic acids (eg 15). Please correct.

-> We have corrected ‘phenolic acids’ to ‘phenolic compounds’ in figure 1.

Line 144:  There is something wrong with the sentence starting on this line. Please correct.

-> We have corrected ‘Becaues’ to ‘because’ in the section 2.3.

Table 1:  Although unsure the opponent thinks  that  the mass error is much lower than stated.  Maybe the authors did not accounted for the mass of one electron (0.00054858  Da). Please consider.

-> Mass error in table 1 was calculated through software, and we used it value. However, we agree with the reviewers. To avoid confusion, it was calculated and corrected in ppm commonly used for High-Resolution Masses. The ppm(parts per millions) was calculated as shown in the figure below.

Ppm : (TM-EM/TM )x 10e6 = ppm error

TM : theoretical m/z value

EM : experimental m/z value

The picture is included in the word-file.

Since the ionization of phenolic compounds is analyzed as negative ion mode, which loses one hydrogen ion, the experimental ion (m/z) was calculated by adding 1.00795, the mass of the hydrogen ion.

Table 2: Please use subscript in molecular formula as per usual.

-> We have corrected this part.

Omit the CIP notation in the tentatively identified compounds. The detection method can not in principle provide such information.

-> checked complete

Figure 2:  The heat map is very useful. Please enlarge all all the text as is it is not easy to read.  Please correct  all the the compound names (eg some start with X). Add units to the value [mg/g dry weight?].

-> We have corrected this part.

Line 190: Please include and properly number the table mentioned in the text with the informations. Table 1 shows completely different informations.

-> We have corrected this part.

Reviewer 3 Report

According to this reviewer, this work can be considered for publication in Molecules after improving certain aspects of the manuscript.

  1. Potential application of developed method should be mentioned deeply in Abstract.
  2. Validation parameters should be introduced in the abstract.
  3. Elements of scientific novelty should be presented in a detailed and convincing manner (in the last paragraph of the Introduction, and shortly in Abstract).
  4. I suggest that a diagram (scheme) presenting the developed analytical procedure used in the study should be added (from sample preparation to final determination).  It would help understand the details of the analytical protocol better, and allow the written description of the procedure to be shortened.
  5. Application of proper quality assurance/quality control (QA/QC) procedures is vital for the measurement results to be treated as a source of reliable analytical information. Consequently, I suggest that a separate section devoted to QA/QC be added to the manuscript. Special attention should be paid to:

- description of the validation procedure for the applied/proposed analytical protocol,

- information on metrological characteristics of the analytical procedure, especially Method Quantitation Limit (MQL) values for the entire procedure (from handling of representative samples to statistical and chemometric evaluation of the data sets obtained), and not only for the analytical techniques used during the analysis of the extracts.

Thus, validation part should be reorganized.

  1. I do not really agree with the Table 2. The concentration of determined analytes should be given; not only the information that the analyte was detected in all samples.. This is not analytical attempt! Remember to report amounts with the corresponding errors (value ± SD). Otherwise, contents are meaningless. Please add where required.
  2. The comparison with other existing methodology should be given. Including the greenes of the developed method. Here you should evaluate the green character. You can use the following metrix:

[1] A. GaÅ‚uszka, Z.M. Migaszewski, P. Konieczka, J. NamieÅ›nik, Analytical Eco-Scale for assessing the greenness of analytical procedures. Trends Anal. Chem., 2012, 37, 61–72.

[2] J. Płotka-Wasylka, A new tool for the evaluation of the analytical procedure: Green Analytical Procedure Index, Talanta, 2018 In press DOI: https://doi.org/10.1016/j.talanta.2018.01.013

  1. Innovative potential of the results obtained should be explained in detail (CONCLUSIONS).

Author Response

We are grateful for your invaluable comments on our manuscript. As per the reviewer’s suggestion, In addition, I have revised and added to reviewers.  Herewith, I have given our response to reviewer comments and revised the manuscript by using shadow text. I hope to get a positive answer from you soon.

  1. Potential application of developed method should be mentioned deeply in Abstract.

-> As per the reviewer's suggestion, the potential application was modified to be mentioned in the abstract.

  1. Validation parameters should be introduced in the abstract.

-> As per the reviewer's suggestion, modified to mention in the abstract the linearity, detection limit, quantitative limit, and precision calculated through method validation.

  1. Elements of scientific novelty should be presented in a detailed and convincing manner (in the last paragraph of the Introduction, and shortly in Abstract).

-> At the end of the introduction, we have added a brief reason for the novelty.

  1. I suggest that a diagram (scheme) presenting the developed analytical procedure used in the study should be added (from sample preparation to final determination). It would help understand the details of the analytical protocol better, and allow the written description of the procedure to be shortened.

-> As per the reviewer's suggestion, we have added a Schematic flow chart.

  1. Application of proper quality assurance/quality control (QA/QC) procedures is vital for the measurement results to be treated as a source of reliable analytical information. Consequently, I suggest that a separate section devoted to QA/QC be added to the manuscript. Special attention should be paid to:

- description of the validation procedure for the applied/proposed analytical protocol,

- information on metrological characteristics of the analytical procedure, especially Method Quantitation Limit (MQL) values for the entire procedure (from handling of representative samples to statistical and chemometric evaluation of the data sets obtained), and not only for the analytical techniques used during the analysis of the extracts.

-> Since there are various specifications for QA/OC, I didn't understand what procedure I was talking about. Moreover, If it is metrological characteristics, does it mean measurement uncertainty? If the measurement uncertainty of our results is necessary, a retest is required. If you provide specific guidelines for reference, we will refer to them.

  1. I do not really agree with the Table 2. The concentration of determined analytes should be given; not only the information that the analyte was detected in all samples.. This is not analytical attempt! Remember to report amounts with the corresponding errors (value ± SD). Otherwise, contents are meaningless. Please add where required.

-> The results in Table 2 show the substances potentially identified for peaks that did not match the information of the standard among the peaks in the entire chromatogram obtained. This is called chemical profiling[1,2] and the method is described in Section 2-3. As a reviewer's opinion, it cannot be quantified and expressed in terms of concentration because it is a substance that does not have a standard substance. However, Q-TOF has already been used in many plant metabolite profiling studies. It also recognizes tentative identification without standard substances [1-6].

The fact that all samples were detected simultaneously may indicate the possibility of some unreported substance present. We also plan to make an accurate confirmation of this in the future. Therefore, our research is also part of the analysis attempt, and we hope that the reviewers will change their minds on this.

Reference

[1] Le, P. M., McCooeye, M., & Windust, A. (2014). Application of UPLC-QTOF-MS in MS E mode for the rapid and precise identification of alkaloids in goldenseal (Hydrastis canadensis). Analytical and bioanalytical chemistry, 406(6), 1739-1749.

[2] Qi, Y., Li, S., Pi, Z., Song, F., Lin, N., Liu, S., & Liu, Z. (2014). Chemical profiling of Wu-tou decoction by UPLC–Q-TOF-MS. Talanta, 118, 21-29.

[3] Pan, X., Welti, R., & Wang, X. (2010). Quantitative analysis of major plant hormones in crude plant extracts by high-performance liquid chromatography–mass spectrometry. Nature protocols, 5(6), 986-992.

[4]Cendrowski, A., Åšcibisz, I., Kieliszek, M., Kolniak-Ostek, J., & Mitek, M. (2017). UPLC-PDA-Q/TOF-MS profile of polyphenolic compounds of liqueurs from Rose petals (Rosa rugosa). Molecules, 22(11), 1832.

[5] Li, Z., Lee, H. W., Liang, X., Liang, D., Wang, Q., Huang, D., & Ong, C. N. (2018). Profiling of phenolic compounds and antioxidant activity of 12 cruciferous vegetables. Molecules, 23(5), 1139.

[6] Xie, S., Shi, Y., Wang, Y., Wu, C., Liu, W., Feng, F., & Xie, N. (2013). Systematic identification and quantification of tetracyclic monoterpenoid oxindole alkaloids in Uncaria rhynchophylla and their fragmentations in Q-TOF-MS spectra. Journal of pharmaceutical and biomedical analysis, 81, 56-64.

  1. The comparison with other existing methodology should be given. Including the greenes of the developed method. Here you should evaluate the green character. You can use the following metrix:

-> As per the reviewer's suggestion, We've added comparison about performing greenness assessments in sections 2-5.

Round 2

Reviewer 3 Report

I accept your response

Author Response

Thank you again for reviewing our manuscript.

Have a happy end of the year.